# Mining the Immunopeptidome for Antigenic Peptides in Cancer

**DOI:** 10.3390/cancers14204968

**Published:** 2022-10-11

**Authors:** Ricardo A. León-Letelier, Hiroyuki Katayama, Sam Hanash

**Affiliations:** Clinical Cancer Prevention, The University of Texas MD Anderson Cancer Center, Houston, TX 77030, USA

**Keywords:** immunopeptidome, PTM, immunotherapy, cancer vaccine

## Abstract

**Simple Summary:**

The immunopeptidome of cancer cells is a treasure trove of neoantigens bound to MHC molecules, thus a great source for mining immunopeptides for immunotherapy applications, including cancer vaccines. Immunopeptides may encompass post-translational modifications that are overlooked by genomic and transcriptomic tools. We review post-translational modifications that have been uncovered, and how this information could be harnessed for cancer vaccines.

**Abstract:**

Although harnessing the immune system for cancer therapy has shown success, response to immunotherapy has been limited. The immunopeptidome of cancer cells presents an opportunity to discover novel antigens for immunotherapy applications. These neoantigens bind to MHC class I and class II molecules. Remarkably, the immunopeptidome encompasses protein post-translation modifications (PTMs) that may not be evident from genome or transcriptome profiling. A case in point is citrullination, which has been demonstrated to induce a strong immune response. In this review, we cover how the immunopeptidome, with a special focus on PTMs, can be utilized to identify cancer-specific antigens for immunotherapeutic applications.

## 1. Introduction

Identifying novel antigens in cancer is highly relevant for immunotherapeutic applications including chimeric antigen receptor (CAR)-T and NK cell, pulsed-dendritic cell therapy, and therapeutic and preventative cancer vaccines [1]. Mass spectrometry provides an important means for deciphering the immunopeptidome repertoire of tumor cells [2]. Whereas much emphasis has been placed on mutations as a source of neoantigens, the occurrence of specific mutations in peptides bound to the major histocompatibility complex (MHC) is quite variable from patient to patient [3,4,5]. Thus, there is a need to identify antigenic peptides that are commonly expressed in a cancer type that are presented through MHC class I for activation of cytotoxic CD8+ T cells [6,7]. Moreover, there is also an emerging interest in immune peptides bound to MHC class II that induce a B cell response [8,9,10].

Since the early days of profiling the immunopeptidome using mass spectrometry (MS) some three decades ago [11], there has been substantial improvement in the overall approach, including the application of machine learning [12,13,14]. The detection and prediction of immunogenic peptides through genomic and transcriptomic data is challenging and overlooks protein aberrations that occur after transcription. These include translational errors, post-translational modifications (PTMs) and peptide splicing that can be uncovered through analysis of the immunopeptidome [15,16,17]. Remarkably, PTMs have been discovered to induce immunogenicity more than their unmodified counterparts. Prior studies by our group have identified citrullination as a source of immunogenicity in cancer [18]. Other notable PTMs include phosphorylation, acetylation, deamination, and glycosylation [19,20,21,22]. However, not all PTMs are stable and presented by MHC, given their enzymatic reversibility as in the case of acetylation [23]. This review covers how interrogating the immunopeptidome can yield novel cancer-specific antigens, with an emphasis on PTMs and their applications.

## 2. The Immunopeptidome as a Source of Different Types of Neoantigens

The immunoediting concept has been critical for our understanding of the mechanisms through which the immune system responds to cancer and how tumor cells can evade the immune response [24]. A key factor in the immune response is the recognition of tumor antigens. T cells, through their TCRs, can interact with the myriad of peptides bound to MHC, sorting out self from non-self. Non-canonical tumor antigens, derived from sequences outside of exons or by alternate protein-processing mechanisms, are of increasing interest for immunotherapy [25]. PTMs are mediated by multiple enzymes, some of which may be dysregulated in tumor cells, rendering them potentially tumor specific. Post-translationally modified proteins undergo processing through the proteasome, resulting in peptides that bind to MHC-I for endogenous proteins or MHC-II for exogenous proteins [26]. Dendritic cells (DCs) are antigen-presenting cells (APCs) in cancer that are essential for T and B cell responses via immunopeptides and native protein presentation, respectively [27,28]. PTMs that are restricted to tumor cells have potential as a source of immunopeptides for immunotherapy.

## 3. Post-Translational Modifications as a Source of Tumor Antigens

Whereas a multitude of PTMs are known to occur, most have not been previously investigated in cancer. Nanoscale liquid chromatography coupled mass spectrometry (nanoLC-MS) has contributed significantly to the identification of PTMs in the immunopeptidome through matching the peptide parent mass (MS1) and the fragment mass (MS2) to sequences in the human genome database, allowing for mass shift due to modified amino acids (e.g., +0.984 Da on Arg) for citrullination; (+97.976 Da on Ser, Thr, and Tyr) for phosphorylation; and (+203.079 Da on Ser and Thr) for O-GlcNAc. In this review, we cover PTMs that have been identified in the immunopeptidome with demonstrated immunogenicity in cancer (Figure 1).

### 3.1. Citrullination

Several studies have explored citrullination as a source of antigenicity in cancer [29,30,31]. Citrullination occurs on arginine residues and is enzymatically driven by peptidyl arginine deiminases (PADI), which are dysregulated in multiple cancer types [18]. The dysregulated citrullination pathway was initially linked to autoimmune diseases, mainly rheumatoid arthritis. More recently, its role in cancer has attracted interest [31,32]. Citrullinated peptides are principally presented by MHC-II, eliciting a CD4+ T cell and B cell response [10,33]. Citrullination in cancer cells occurs as a result of cellular stress, exemplified by autophagy, nutrient starvation, and hypoxia [31,34,35,36], which induces PADI expression. The PADI family consists of five members, with PADI2 and PADI4 being predominantly expressed in cancer [37].

In a recent study, we found PADI2 to be highly expressed in several cancer types, notably in triple-negative breast cancer (TNBC) [18]. PADI2 expression was correlated with accumulation of citrullinated proteins and, with MHC-II-bound citrullinated peptides, and with a B cell immune response. PADI2 is also overexpressed in HER2+ breast cancer, hepatocellular carcinoma, esophageal cancer, gastric adenocarcinoma, and castration-resistant prostate cancer [38,39,40]. PADI2 is variably expressed in colorectal cancer (CRC) [41,42]. It is intriguing that low PADI2 expression seems to correlate with poor prognosis, possibly due to a lack of immunogenic citrullinated peptides [41,43]. Overexpression of PADI2 in skin tumors is associated with elevated inflammatory cell infiltration [31,44,45]. CRC with high PADI2 and PADI4 expression is associated with increased overall survival [46]. PADI4 expression in benign tumors and non-tumor inflamed tissues was found to be restricted to malignant tumors in gastric, liver, and ovarian cancers [47,48,49,50,51].

Citrullination has been investigated in relation to tumor biology and metastasis and as a source of cancer biomarkers [31]. For applications to immunotherapy usage, some citrullinated proteins have been tested as antigen candidates for therapeutic cancer vaccines. Among several hundred citrullinated proteins [52], α-enolase (ENO1), vimentin (VIM), nucleophosmin (NPM1), matrix metalloproteinase-21 (MMP21), cytochrome p450 (Cp450), and glutamate receptor ionotropic (GRI) citrullinated peptides have been selected for immunization for melanoma, lung, pancreas, and ovarian cancers [29,30,33,53,54,55].

### 3.2. Phosphorylation

The majority of phosphorylation research in cancer is focused on signaling pathways [56]. However, there is interest in phosphorylation as a neoantigen target. The most abundant of the enzymatically modified cancer proteins are represented by phosphoproteins, resulting from dysregulation of kinase-mediated signaling pathways triggering the synthesis of cancer-associated phosphopeptides [57]. Importantly, MHC-I and MHC-II present these phosphorylated peptides in cancer cells [58,59,60]. Moreover, there are reports of tumor-specific phosphorylated peptides stimulating CD8 and CD4 T cells and even phosphopeptide-specific T cells in cancer [61,62,63]. There is also evidence of a B cell response against phosphorylated proteins [64]. These findings elicit interest in phosphopeptides as a source of cancer antigens [65,66].

Two phosphopeptides are currently being tested in melanoma patients, one derived from the insulin receptor substrate 2 (IRS2) and the other from breast cancer antiestrogen resistance 3 (BCAR3) (NCT01846143). In CRC, 120 phosphopeptides were identified, some of which were tumor restricted. TILs from CRC patients recognized three of the identified phosphopeptides from IRS2, tensin 3 (TNS3), and selenoprotein H (SELH) [67]. Other phosphopeptides with potential utility as immunotherapeutic targets are derived from beta-catenin and CDC25b [58,62]. Approaches to enrich for phosphopeptides in the immunopeptidome include the use of immobilized metal affinity chromatography and titanium dioxide nanoparticles [68,69,70]. In development, is a predictor of interactions between MHC-I and phosphopeptides. [71].

### 3.3. Glycosylation

Protein glycosylation occurs in the endoplasmatic reticulum and Golgi apparatus and is commonly associated with secreted and extracellular membrane proteins. Considering that there are several types of glycans that could be covalently conjugated to proteins, there are a myriad of possible combinations. O-linked-N-Acetylglucosamine (O-GlcNAc) plays an important role in signal transduction, transcription, cell division, metabolism, and the cytoskeleton in cancer cells [72]. Given that glycosylation creates novel epitopes, its occurrence in peptides presented by MHC has been of interest as a source of targets for immunotherapy [73].

An important source of glycopeptides is represented by mucins (MUC), which are highly glycosylated in cancer. They have been utilized as a study model for carbohydrates’ immunogenicity [74,75]. Tumor cells overexpress MUC1 and MUC4 on the surface, both of which exhibit altered glycosylation, potentially representing tumor-specific targets. Common antigens expressed in cancer mucins are the O-glycans Tn and T antigens, which have been utilized as vaccines. Several vaccine strategies using glycopeptides from mucins in combination with adjuvant have been utilized to induce an anti-tumoral response in vitro and in vivo. These strategies are being put to the test for different cancer types, including breast, prostate, lung, pancreatic, and renal cancer, as well as melanoma and lymphoma [76]. The glycosylation pattern of MUC proteins has proven to be a useful target for single-domain antibodies overcoming tumor growth, invasion, and metastasis in a mouse model [77].

Other glycosylated immunopeptides have also been investigated as vaccines, tumor-selective antibodies, CAR T cells, nanoparticles, and DC therapy [78,79,80,81,82,83,84,85,86,87,88]. Critical to this effect is knowledge of the structure of glycoproteins for their synthesis and antibody binding properties and their surface localization and occurrence in the immunopeptidome [89,90,91,92,93,94,95,96,97,98,99,100]. Interestingly, MHC-I also undergoes glycosylation, which must be taken into account for antigen presentation [101]. Additionally, of interest is the finding that deaminated MHC-I-bound peptides are derived from glycopeptides, which has relevance to antigen presentation to T cells [97]. At present, there is a surge of interest in glycopeptides presented by MHC-II on DCs, resulting from a deeper understanding of how glycosylated proteins are presented [102]. These advances and diversified strategies around the glycoproteogenome and its immunopeptidome are promising, not only for glycopeptides but also for various peptide PTMs in cancer that could be utilized as tumor-specific targets.

## 4. Peptide PTMs as a Source of Cancer Vaccines

Taking into account that the immunopeptidome represents the whole spectrum of peptides presented in a cell, there is a need to identify the most promising cancer targets. Thus, there is a need to determine the structure of an MHC-bound peptide and its level of expression for vaccine development. There are numerous ongoing clinical trials utilizing different antigens and adjuvants as therapeutic cancer vaccines. Focusing on peptides with PTMs as vaccines, promising findings have resulted from the use of citrullinated peptides (Table 1). A citrullinated VIM peptide has been utilized as an antigen in combination with an adjuvant induced IFN-γ and granzyme B-secreting CD4+ T cells. Citrullinated VIM-specific Th1 cells induced by the vaccine had a potent antitumor response against established skin and lung tumors, as well as a long-term memory response [30]. Similarly, a citrullinated ENO1 peptide-based vaccine elicited a potent citrulline-specific Th1 cell response in pancreatic, skin, and lung cancers [29]. Additionally, ENO1 is commonly overexpressed in different tumor types, including melanoma, pancreatic, breast, and lung cancer, thus citrullinated ENO1 peptides are plausible antigens for a wide cancer spectrum [18,29,55]. Furthermore, the combination of citrullinated VIM and ENO1 peptides in a vaccine, designated Modi-1, induced a significant antitumoral response in a mouse model of ovarian cancer. Importantly, a substantial citrulline-specific T cell response was observed in more than half of ovarian cancer patients [103]. Moreover, analysis of the melanoma immunopeptidome led to the identification of MHC-II-bound citrullinated peptides are derived from MMP21, Cp450, and GRI proteins [33]. A combination of these citrullinated peptides did not induce a greater antitumoral response than citrullinated MMP21 and GRI peptides individually, pointing to the potential of a reduced response with multiple peptides with different MHC-II binding specificities [33]. Another source of citrullinated peptides is the NPM protein. Vaccination with a PADI2-mediated-citrullinated NPM peptide induced an antitumoral response which was therapeutic, increasing survival and resulting in protection against a second tumor challenge in melanoma and lung cancer mouse models. Interestingly, PADI4-mediated-citrullination of NPM peptide did not elicit a citrulline-specific Th1 response, in contrast to PADI2-mediated-citrullination [53]. The CD4 responses observed may result from binding of citrullinated peptides primarily by MHC-II [18] in HLA-DP4 and HLA-DR4 transgenic mice. Nevertheless, the vaccine-induced CD4 response was sufficient to inhibit tumor progression, indicating the effectiveness of responses that do not involve CD8+ T cells [10].

There is a more limited number of studies utilizing phosphorylation as a PTM for peptide vaccines, although immunopeptidome analysis has pointed to a substantial number of phosphorylated peptides. Immunopeptidome analysis of melanoma, ovarian carcinoma, B lymphoblastoid, and leukemia resulted in the identification of a large number of phosphopeptides that were cancer specific with CD8 T cell antigen specificity in patients [58,63]. Some of the identified phosphopeptides were derived from ISR2, BCAR, TNS2, SELH, CDC25b, and beta-catenin [58,62,63], concordant with the identification of phosphorylated ISR2, TNS2, and SELH peptides in the colorectal cancer immunopeptidome [67]. At present, phosphopeptides from ISR2 and BCAR are being explored as a cancer vaccine for melanoma patients [104]. The phase I trial confirmed that the vaccines using these peptides are safe and capable of inducing an immune response, justifying future studies for their use as vaccines (NCT01846143).

As for glycopeptide-based cancer vaccines, an initial source was the glycosylated MUC protein displaying the Tn antigen. Immunization of mice with a desialylated ovine MUC with substantial representation of the Tn antigen elicited primarily a CD4+ T cell response specific to the Tn antigen. Conversely, immunization with a deglycosylated MUC did not induce an immune response [73]. The induction of an immune response specifically against the PTM protein suggests that glycosylation may be a useful source of cancer-specific antigens given the findings of aberrant glycosylation in many cancers, notably breast cancer [78,106]. A case in point is a fully synthetic cancer vaccine, a dendrimeric multiple antigenic glycopeptide displaying a trimer of Tn antigens (MAG-Tn3) associated with a promiscuous CD4 epitope, the tetanus toxoid-derived P2 peptide, that has been shown to induce an antitumoral Tn-specific T cell response in monkeys [78]. This MAG-Tn3 vaccine has been used in a phase I clinical trial for high-risk relapsed breast cancer patients (NCT02364492). Another cancer vaccine in clinical trial is based on MUC1 bearing Tn antigens (Tn-MUC1) pulsed with autologous DCs [81]. This phase I/II clinical trial follows the same strategy used in rhesus macaques, which resulted in five out of seven castrate-resistant prostate cancer patients having a CD4 and/or CD8 response (NCT00852007).

Other studies involving MUC1 glycopeptide vaccines induced a monoclonal IgG specific to mammary tumors [80]. Likewise, a synthetic MUC1 glycopeptide linked to a B-cell and a T-cell epitope together with poly I:C as an adjuvant elicited significant IgG titers against tumor-associated MUC1 expressed in breast cancer cells [82]. Recently, a novel approach was developed using a MUC1 glycopeptide that consists of a fluorinated nanoliposomal vaccine that is self-adjuvanated. This novel vaccine induced a high level of antigen-specific IgG in mice [107]. These glycopeptide-based vaccines were designed to induce primarily a humoral response. However, there are cancer vaccines using MUC1 glycopeptides that bind to MHC-I, inducing a cytotoxic CD8 response observed in healthy donors as well as in breast cancer patients [108]. There are also some promising glycopeptide candidates identified from leukemia MHC-I immunopeptidomes, five of which have been associated with a memory T cell response in healthy subjects [96]. Likewise, MUC4 glycopeptide candidates have been identified for pancreatic cancer immunotherapy [85,109]. Given the complexity of glycan modifications, there has been a surge of various approaches to identify and develop glycopeptides as vaccines, as reviewed above, including the use of glyco-antigen microarrays to investigate immune responses to cancer vaccines [79,91]. Another development is the use of an antigen delivery system based on gold nanoparticles with Dectin-1 to target DC, conjugated with MHC-II glycopeptides. This gold nanoparticle glycopeptide vaccine elicited a strong humoral and cellular immune response in mice [84]. In all, much progress has been made in the identification of glycopeptides and their structural and other properties to enhance their effectiveness as cancer vaccines [92,93,95,98].

## 5. Conclusions

It is evident that the potential of harnessing the immunopeptidome with its PTM peptides for cancer therapy and vaccines is quite substantial. Reliance on PTM modifications in tumor antigens further enhances the specificity of the epitopes and their restricted expression to cancer. Such PTMs do not occur in the thymus, resulting in a lack of negative selection for corresponding T cells. Although this review covered citrullination, phosphorylation, and glycosylation PTMs, there are numerous other modifications, some consisting of fusion peptides resulting from aberrant proteasomal function, known as proteasomal splicing. These spliced peptides result from the fusion of two unrelated fragments presented by MHC in cancer cells but may not be cancer specific [15,110]. An interesting development is harnessing the immunopeptidome to generate personalized oncolytic cancer vaccines, as demonstrated using a murine colon cancer model. This impressive immunopeptidomic-based pipeline harnesses the entire MHC-bound peptidome to develop an oncolytic cancer vaccine coated with tumor antigen peptides as a tool for immunotherapy [111]. The immunopeptidome also provides a basis for CAR T cell therapy, [83,112,113,114], targeting glycopeptides with promising results [86,115,116]. An example is the development of a peptide-centric CAR T cell using the immunopeptidome of neuroblastoma. Remarkably, computational modeling predicted that this peptide-centric CAR T cell was capable of recognizing peptides with different MHC-I polymorphisms, resulting in a strong and specific killing of neuroplastoma cells and complete tumor regression in mice [105,117].

Another promising avenue for immunopeptide-based immunotherapies is in combination with immune checkpoint inhibitors (ICI) with the potential for synergism as with anti-PD-1 immunotherapy [118]. Taking into account that ICI is mostly effective in the presence of tumor infiltrating lymphocytes, a prior immunization that can efficiently induce T cell tumor migration would enhance the efficiency of ICI therapy [119,120]. All things considered, the immunopeptidome field has crucial relevance for cancer interception.

## Figures and Tables

**Figure 1 cancers-14-04968-f001:**
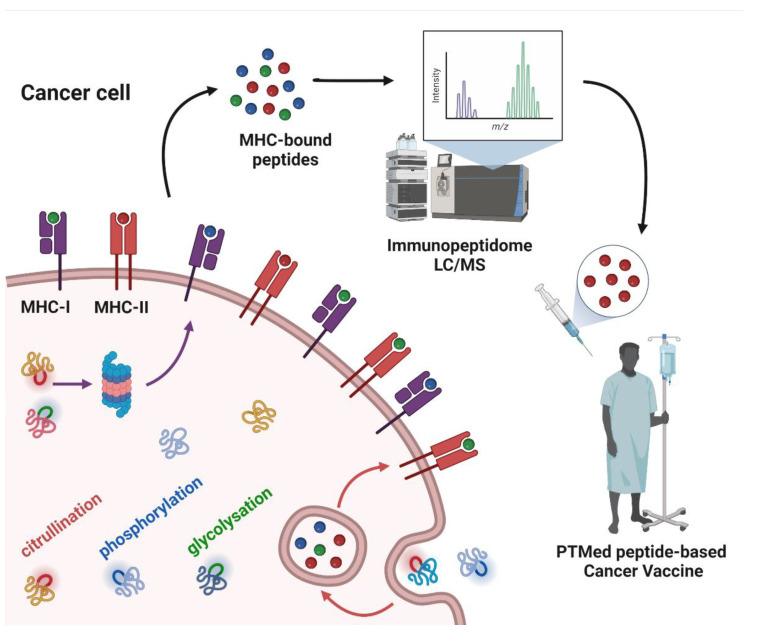
Post-translationally modified peptide-based cancer vaccine workflow. The figure depicts cancer cell antigen processing of intracellular and extracellular proteins, subsequently as peptides bound to MHC-I or MHC-II. Some of the proteins have PTMs in their structure, which are sketched in colors (citrullination: red; phosphorylation: blue; glycosylation: green) as well as in MHC-bound peptides. The MHC-bound peptides are identified by means of liquid chromatography-mass spectrometry (LC/MS), to derive the cancer cell immunopeptidome. From the immunopeptidome data, peptides with PTMs can be selected as antigens for cancer vaccines.

**Table 1 cancers-14-04968-t001:** Summary of the post-translational modified peptides used in immunotherapy.

Post-Translational Modification	Protein	Cancer Type	Immunotherapy	MHC Class	Reference
Citrullination	ENO1	SKCM, PAAD, LUAD, OV	Vaccine	II	[18,29,55,103]
	VIM	SKCM, LUAD, PAAD, OV	Vaccine	II	[30,103]
	MMP21	SKCM	Vaccine	II	[33]
	GRI	SKCM	Vaccine	II	[33]
	Cp450	SKCM	Vaccine	II	[33]
	NPM	SKCM, LUAD	Vaccine	II	[53]
Phosphorylation	ISR2	SKCM	Vaccine, ACT	I	[62,104]
	BCAR	SKCM	Vaccine	I	[104]
	CDC25b	SKCM	ACT	I	[62]
Glycosylation	MUC1	BRCA, PRAD	Vaccine, DCTher	I, II	[77,80,81,82]
	MUC4	NA	Vaccine	II	[85]
	PHOX2B	Neuroblastoma	CAR T cell	I	[105]

ACT: Adoptive Cell Therapy; DCTher: Dendritic Cell Therapy; ENO1: α-enolase 1; VIM: vimentin; MMP21: Matrix Metalloproteinase-21; GRI: Glutamate Receptor Ionotropic; NPM: Nucleophosmin; ISR2: Insulin Receptor Substrate 2; BCAR: Breast Cancer Antiestrogen Resistance 3; SKCM: Skin Cutaneous Melanoma; PAAD: Pancreatic Adenocarcinoma; PRAD: Prostate Adenocarcinoma; BRCA: Breast Invasive Carcinoma; LUAD: Lung Adenocarcinoma; OV: Ovarian Serous Cystadenocarcinoma; NA; Not Applicable.

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
