# Peer review of "Mining the Immunopeptidome for Antigenic Peptides in Cancer"

_cancers, 2022, doi:10.3390/cancers14204968_

Round 1

Reviewer 1 Report

The review written by Letilier et.al covers how immunopeptidome can be utilized to identify cancer specific antigens which can further be used for immunotherapeutic application. The review is well written covering the area is Citrullination, Phosphorylation and Glycosylation and covered various aspects where PTM peptides can be used as a source of vaccine

Just a few errors in reference was found where in some reference volume or page number was missing, for instance reference number 28, 60, 64, 69 and 105

Please cross verify the year of publication for reference number 60, 64 and 69

Please check reference format and unify all as some are written First author followed by et.al, but for few reference all authors are listed

Author Response

Thank you for the comments, specially about the errors in the references. We have corrected references 28, 60, 64 and 105, double checking the other references as well. Also, we verified the year of publication for references 60, 64 and 69. Lastly, we unified the format showing only the first author followed by et al. when there are more than 2 authors in the article.

Reviewer 2 Report

This review mainly introduces several types of post-translational modifications as the source of tumor antigens, and summarizes the latest research background in this area, which is beneficial to provide reference for other people's research. The advantages are that the content is broadly relevant to the topic, the research background is clearly presented, the research methodology is robust and used appropriately, the discussion in the literature will largely be found in a broad context, and largely explained and usefully speculative and verbose. A balance between the descriptions of the results, the results section is compared to other similar published studies, the results or proposed methods have some potential broader applicability or relevance, and the language is fluent and logical, and the cited articles have a certain authority and reference value. The disadvantage is that the author's thinking in this field is not deep enough, and his own insights are less.

Author Response

We appreciate the feedback from the reviewer and the concerns raised. To address them we elaborated further on findings that we considered to be most novel and interesting for the scope of the Review, in that way adding more insight to our work. This is provided in related sections from lines 67-72, 183-186, and 288-292.

Reviewer 3 Report

The authors of the manuscript, “Mining the Immunopeptidome for Antigenic Peptides in Cancer”, have discussed the mainly the post translational modifications of the peptides that can generate tumor antigens and elicit an immune response in cancer patients. These modifications can not be predicted from the mutations of the genome or from transcriptome profiling. The authors have further the discussed the importance of utilizing these post translational modified peptides alone or in combination with other treatments for cancer immunotherapy. The manuscript is well written, and I have few suggestions that can significantly improve the quality of the manuscript as below.

Comments

 -In the abstract, the authors of the manuscript have mentioned only neoantigens that bind to MHC-I and have excluded neoantigens that bind to MHC-II but in the rest of the manuscript the authors have also focused on for example citrullinated peptides, phosphorylated peptides principally presented by MHC-II and eliciting a CD4 or B cell response. The authors should rewrite the abstract to include and summarize the contents of the manuscript.

-The authors have beautifully discussed post translational modifications as a source of tumor antigens and it would be nice if authors can add few lines and give a glimpse of about how the immunopeptidome for these antigens such as glycosylated, phosphorylated, or other antigens have been harnessed and analyzed by mass spectrometry.

Author Response

Thank you for the suggestions, clearly they  improve the quality of the review. For the first comment we added MHC-II in the abstract (line 11). For the second comment we explained how PTMs are analyzed by mass spectrometry (line 67-72), and how peptide PTMs are used in cancer vaccines (Section 4. Peptide PTMs as a source of cancer vaccines)